# LEARNING RATE RE-SCHEDULING FOR ADAGRAD IN TRAINING DEEP NEURAL NETWORKS

## ABSTRACT

The adaptive learning rate optimization algorithms have made a great improvement in the training of Deep Neural Networks (DNNs). It has been proved that adaptive learning rate methods can significantly improve training processing and can be adopted into various tasks. AdaGrad, As the first adaptive learning rate optimizer, usually performs worse than the following optimizers, such as Adam, RAdam, Adabelief, *etc*. There are mainly two reasons: the first is that the stepsize for these optimizers is bounded so that the training is more stable, and the second is that they can use the decoupled weight decay regularization to improve their generalization performance. However, for AdaGrad, the updating delta constantly decreases to zero. Consequently, the weights will change very slowly with the number of training iterations increasing. Meanwhile, it also makes the decoupled weight decay regularization perform unfavorably in AdaGrad. We find that there is a big mistake when using AdaGrad in training DNNs. For other optimizers (*e.g.*, Adam), they prove the regret-bound theorem with learning rate schedule $\frac{1}{\sqrt{T}}$, but in practice, they usually use more advanced learning rate schedule for training DNNs, such as step-wise decay schedule and cosine decay schedule. However, for AdaGrad, the algorithm implicitly contains a learning rate schedule $\frac{1}{\sqrt{T}}$, but in practice, most people directly add another learning rate schedule for AdaGrad. Such two learning rate schedules will largely drop its performance in training DNNs. So in this work, we propose a Learning Rate Re-scheduling (LRR) method for AdaGrad to drop the implicit learning rate $\frac{1}{\sqrt{T}}$, which can largely improve AdaGrad and make decoupled weight decay regularization perform well. The proposed LRR method can also be applied to other AdaGrad-type algorithms (*i.e.*, Shampoo). Comprehensive experiments indicate the effectiveness of the proposed LRR method. The source code will be made publicly available.

## 1 INTRODUCTION

As a basic optimization algorithm in deep learning, Stochastic gradient descent (SGD) Robbins & Monro (1951) has achieved remarkable performance in training Deep Neural Networks (DNNs). By using the back-propagation algorithm, the gradient of parameters in a Deep Neural Network (DNN) can be easily obtained. SGD updates the weight along the opposite gradient direction in each iteration. A significant improvement on SGD is to compute the momentum of gradient Qian (1999) (SGDM) so that it can speed up SGD in the relevant direction and reduce oscillations. To achieve a lower regret bound, *et al*. Duchi et al. (2011) proposed the famous AdaGrad algorithm. Instead of using a uniform learning rate for all parameters, AdaGrad assigns an adaptive learning rate for each parameter independently. Specifically, AdaGrad uses the sum of the historical second-order statistic of the gradient to compute the adaptive learning rate, *i.e.*, $(\sum_{t=1}^{T} \mathbf{g}_t \odot \mathbf{g}_t)^{\odot - \frac{1}{2}}$.

However, with the AdaGrad optimizer, the effective learning rate will decrease during training. As a result, it usually does not perform well in many real applications. To solve this problem, RMSProp Tieleman & Hinton (2012) suggested using the Exponential Moving Average (EMA) of the second-order statistic of the gradient for computing the adaptive learning rate to replace the sum in AdaGrad. Meanwhile, Adam Kingma & Ba (2014) further combines the adaptive learning rate strategy of RMSProp with the momentum of the gradient. Other Adam-type optimizers are also proposed in the following research, such as RAdam Liu et al. (2019a), Adabelief Zhuang et al. (2020),

Ranger Liu et al. (2019b); Zhang et al. (2019); Yong et al. (2020) and so on. More importantly, the weight decoupled strategy Loshchilov & Hutter (2017), which modifies the weight decay approach of Adam, can be adopted to improve the final generalization performance. The weight-decoupled strategy has become the standard way to introduce weight decay in the Adam-type optimizers. It has been widely verified in many works Loshchilov & Hutter (2017); Vaswani et al. (2017); He et al. (2022); Li et al. (2022), that the decoupled weight decay strategy usually keeps a better final generalization for Adam optimzer.

For AdaGrad, its unsatisfactory performance mainly comes from two reasons. The first is that the effective learning rate decreases during training. And for some parameters, the responding learning rate may decrease too fast so that they are not fully optimized. After several iterations, such parameters would not change. The second reason is that there are no proper weight decay methods to improve the generalization performance of AdaGrad. The original L2 regularization weight decay method is usually used in AdaGrad, which has been proven ineffectiveness in Adaptive learning rate methods Loshchilov & Hutter (2017). Nevertheless, the decoupled weight decay usually performs very unfavorably in the AdaGrad-type optimizers. It has a negative impact on the optimization of the loss function. The reason is that the effective learning rate decreases during training with AdaGrad so that the decoupled weight decay will dominate the updating direction after certain iterations. Consequently, the weights will be too small to lead to an unfavorable performance.

Actually, we find there is an incorrect way to utilize AdaGrad in training DNNs, which is the biggest reason for the bad performance of AdaGrad. For the Adam-type optimizers (*e.g.*, Adam, RAdam and Adabelief), they prove the regret-bound theorem with learning rate schedule $\frac{1}{\sqrt{T}}$, but in practice, they usually use more advanced learning rate schedules instead of it for training DNNs, such as step-wise decay schedule, cosine decay schedule, and warm-up strategy. These learning rate schedules usually keep better performance in training DNNs. However, for AdaGrad, people usually directly utilize such learning rate schedules for training DNNs. There is a very big mistake in that the AdaGrad algorithm implicitly contains a learning rate schedule $\frac{1}{\sqrt{T}}$. As a consequence, there will be two learning rate schedules: One is the implicit learning rate schedule $\frac{1}{\sqrt{T}}$ and the other is the practical learning rate schedule we used, which can be step-wise decay schedule, cosine decay schedule and so on. Such two learning rate schedules will make the learning rate decrease so fast that the effective learning rate will be too small.

To address this problem in AdaGrad, we propose a very simple learning rate re-scheduling method in training with AdaGrad. It multiplies a constant $\sqrt{T}$ on the global learning rate to offset the original implicit learning rate schedule and then adopts another learning rate schedule. With the proposed learning rate re-scheduling (LRR) method, the performance of AdaGrad can largely improve in training DNNs. Moreover, the decoupled weight decay can also be easily utilized in the modified AdaGrad, which can further gain the final generalization performance. The main contributions of this paper are highlighted as follows:

- We first point out the incorrect utilization of AdaGrad optimizer in training DNNs, which is the two learning rate schedulers in practice. Such a mistake is the major reason that makes AdaGrad an unsatisfactory performance.

- We propose a very simple and effective method, learning rate re-scheduling (LRR), to improve AdaGrad, which can largely improve its optimization on DNNs. The decoupled weight decay regularization is also introduced to further improve the generalization performance.

- We extend the proposed LRR method on another AdaGrad-type optimizer, Shampoo Gupta et al. (2018), which also achieves a favorable performance gain.

Finally, we perform comprehensive experiments on image classification tasks on CIFAR100/CIFAR10 and ImageNet to show the effectiveness of the proposed learning rate re-scheduling for AdaGrad-type optimizers on training DNNs.

## 2 RELATED WORKS

**Adaptive Learning Rate Optimizers:** In spite of a uniform learning rate for all parameters, Duchi et al. (2011) first proposed the AdaGrad method, which adopts an adaptive learning rate for each parameter. It can be proved that AdaGrad can achieve a lower regret bound than SGD. RMSProp

was proposed by Tieleman & Hinton (2012), which introduces the exponential moving average to replace the second-order statistics of the gradient. Adam Kingma & Ba (2014) further uses the momentum of the gradient to make training more stable. It can be shown that the updating value of Adam can be bounded in each iteration. RAdam Liu et al. (2019a) improves the warm-up step of Adam by controlling the variance of the adaptive learning rate. Adabelief Zhuang et al. (2020) adjusts the learning rate by the variance in observed gradients. The adaptive learning rate methods can outperform SGDM in many applications, including image low-level vision Zhang et al. (2017); Zheng et al. (2021), natural language processing Li et al. (2022); Vaswani et al. (2017), and the optimization of Transformer Backbones Vaswani et al. (2017); He et al. (2022).

**Preconditioned Gradient Descent Methods:** In order to achieve a lower regret-bound than AdaGrad, Duchi et al. (2011) also provided a full-matrix preconditioned gradient descent (PGD) method that uses the matrix $\boldsymbol{H}_T = (\sum_{t=1}^{T} \boldsymbol{g}_t \boldsymbol{g}_t^\top)^{\frac{1}{2}}$ to modify the gradient. The adaptive learning rate methods only consider the diagonal elements of $\boldsymbol{H}_T$. Because of the high computation and memory costs for this full-matrix preconditioned gradient descent, various works try to make it practical in training DNNs. Agarwal et al. (2019) proposed to store only the gradients of recent iterations to efficiently approach $\boldsymbol{H}_T$ by low-rank computation tricks. Shampoo Gupta et al. (2018) and AdaBK Yong et al. (2023) were proposed to utilize the Kronecker products to reduce the dimension of full-matrix preconditioners and make it more efficient in the optimization of DNNs. However, these AdaGrad-type optimizers also have the same drawbacks as AdaGrad, which lead to them not being widely used.

**Weight Regularization:** L2 regularization weight decay Krogh & Hertz (1991) was proposed to improve the generalization performance of SGD optimizers. It adds the gradient of L2 regularization into the gradient of weight and then implements the optimizer updating step. However, for the adaptive learning rate optimizers, this weight decay method usually performs unsatisfactorily. Ilya *et al.* Loshchilov & Hutter (2017) proposed to use a decoupled weight decay to replace the L2 regularization weight decay. It directly adds a weight decay term into the finally updated weight in the implementation of the optimizer step. It has been the most common choice of weight decay in Adam-type optimizers, because of the large improvements in generalization. Other regularizations on weight such as Weight normalization (WN) Salimans & Kingma (2016) and Weight standardization (WS) Qiao et al. (2019) use hard constraints on weight, which can also boost the performance. Weight decay remains the most simple and effective weight regularization method.

**Learning Rate Scheduler:** A constant learning rate commonly cannot achieve satisfactory performance in training DNNs. then gradually reducing the learning rate in accordance with a scheduler usually performs well. The most widely used Learning Rate scheduler is the step-wise decay scheduler, which reduces the learning rate by a certain amount every several epochs. It has been widely use in many works He et al. (2016); Ren et al. (2015); Luong et al. (2015). Loshchilov & Hutter (2016) proposed a cosine decay schedule by changing the learning rate with a Cosine function, which can usually perform better than step-wise decay scheduler. Goyal et al. (2017) also constituted an important ingredient in training deep networks, the warmup learning rate method, which involves increasing the learning rate to a large value over a certain number of training iterations followed by another learning rate scheduler. By the way, for the AdaGrad-type optimizers, there usually is an implicit learning rate scheduler $\frac{1}{\sqrt{T}}$, which is usually ignored in practice.

## 3 METHODOLOGY

### 3.1 PROBLEMS IN THE ADAGRAD ALGORITHM

The motivation of the AdaGrad Algorithm is to use different learning rates for each parameter based on iteration. The reason for the utilization of different learning rates is that the learning rate for sparse feature parameters needs to be higher compared to the dense features parameter because the frequency of occurrence of sparse features is lower. It can be explained by an online mirror descent with an adaptive time-dependent regularization. Suppose we have obtained the gradient $\boldsymbol{g}_T = \nabla f_T(\boldsymbol{w}_T)$ in the $T$-th iteration, where $\boldsymbol{w}_T \in \mathbb{R}^d$, then given a positive semidefinite (PSD) matrix $\boldsymbol{H}_T$, the parameters are updated by optimizing the following problem on weight $\boldsymbol{w}$:

$$\boldsymbol{w}_T = \arg\min_{\boldsymbol{w}} \alpha \boldsymbol{g}_T^\top \boldsymbol{w} + \frac{1}{2}||\boldsymbol{w} - \boldsymbol{w}_{T-1}||_{\boldsymbol{H}_T}^2. \tag{1}$$

The solution to the above problem is

$$\boldsymbol{w}_T = \boldsymbol{w}_{T-1} - \alpha \boldsymbol{H}_T^{-1} \boldsymbol{g}_T. \tag{2}$$

which is a preconditioned gradient descent step, where $\alpha$ is the learning rate. Different choices of $\boldsymbol{H}_T$ lead to different optimization algorithms. Duchi et al. (2011) proposed to use a diagonal matrix, which is

$$\boldsymbol{H}_T = \text{Diag}\big(\boldsymbol{h}_T^{\odot \frac{1}{2}}\big), \quad \boldsymbol{h}_T = \sum\nolimits_{t=1}^{T} \boldsymbol{g}_t \odot \boldsymbol{g}_t \tag{3}$$

where $\boldsymbol{A} \odot \boldsymbol{B}$ and $\boldsymbol{A}^{\odot \alpha}$ are the element-wise matrix product and element-wise power operation, respectively. It can be shown that such a choice can provide a lower regret-bound than simple SGD.

However, it can be easy to prove that $\boldsymbol{h}_{i,T+1} \geq \boldsymbol{h}_{i,T}$, for any $i$. It accumulates the sum of past gradients and current gradient, which leads to the effective learning rate $\boldsymbol{h}_{i,T}^{-\frac{1}{2}}$ will monotonically decrease during the training process. And for some parameters, the responding learning rate may decrease too fast and very close to zero, so that they cannot be fully optimized. After several iterations, such parameters would not change. This causes updates to stall early and training to end early.

Another serious problem is that there are no proper weight decay methods to improve the generalization performance of AdaGrad. Loshchilov & Hutter (2017) found that the original L2 regularization weight decay usually cannot work well in the adaptive learning rate methods. So they proposed the decoupled weight decay to replace the L2 regularization weight decay, which directly adds a weight decay term into the finally updated weight in the implementation of the optimizer step. It has been proved that such a weight decay method can improve the generalization performance of many adaptive learning rate methods, including Adam Kingma & Ba (2014); Loshchilov & Hutter (2017), RAdam Liu et al. (2019a), Adabelief Zhuang et al. (2020), *etc*. Nevertheless, the decoupled weight decay usually has a negative impact on the training process of AdaGrad. The updating formulations of AdaGrad with L2 regularization weight decay and decoupled weight decay are

$$\text{L2 regularization weight decay:} \quad \hat{\boldsymbol{g}}_T = \boldsymbol{g}_T + \lambda \boldsymbol{w}_{T-1} \quad \hat{\boldsymbol{h}}_T = \sum\nolimits_{t=1}^{T} \hat{\boldsymbol{g}}_t \odot \hat{\boldsymbol{g}}_t,$$

$$\boldsymbol{w}_T = \boldsymbol{w}_{T-1} - \alpha \hat{\boldsymbol{h}}_T^{\odot -\frac{1}{2}} \odot \hat{\boldsymbol{g}}_T, \tag{4}$$

$$\text{Decoupled weight decay:} \quad \boldsymbol{w}_T = \boldsymbol{w}_{T-1} - \alpha \big(\boldsymbol{h}_T^{\odot -\frac{1}{2}} \odot \boldsymbol{g}_T + \lambda \boldsymbol{w}_{T-1}\big),$$

where $\lambda$ is a hyper-parameter to control the strength of the weight decay. We can find that for the decoupled weight decay, the updating step is controlled by two terms: the stepsize $\boldsymbol{h}_T^{\odot -\frac{1}{2}} \odot \boldsymbol{g}_T$ of AdaGard and the decoupled weight decay term $\lambda \boldsymbol{w}_T$. Nevertheless, the first term may constantly decrease during training because of the monotonically decreasing effective learning rate $\boldsymbol{h}_T^{\odot -\frac{1}{2}}$, and the second term is usually stable in training. As a result, the decoupled weight decay term will dominate the updating direction after certain iterations. Consequently, the weights will be too small to lead to an unfavorable impact on the optimization of the original loss function. Therefore, in practice, we find the training loss will be very large with decoupled weight decay in AdaGrad (as shown in Figure 1 ).

## 3.2 IMPLICIT LEARNING RATE SCHEDULER IN ADAGRAD

As the most common theoretical tool, the online convex optimization framework Shalev-Shwartz et al. (2012); Hazan et al. (2016) tries to minimize the regret to analyze the convergence of an optimization algorithm. For a unknown sequence of convex loss functions $f_1(\mathbf{w}), f_2(\mathbf{w}), ..., f_T(\mathbf{w})$, the regret on $T$-th iteration is

$$R(T) = \sum_{t=1}^{T} \left( f_t(\mathbf{w}_t) - f_t(\mathbf{w}^*) \right), \tag{5}$$

where $\mathbf{w}^* = \arg\min_{\mathbf{w}} \sum_{t=1}^{T} f_t(\mathbf{w})$. Stochastic convex optimization can be viewed as a special case of online convex optimization.

Here we find the inconsistency between the learning rate schedulers in Regret-bound analysis and in practice for Adam-type optimizers. For instance, for Adam, the Regret-bound analysis in the original paper Kingma & Ba (2014) makes some mistakes. Reddi et al. (2019) finished the Regret-bound Theorems with an AMSGrad operation for Adam. All the Regret-bound Theorems of Adam assume the learning rate in $t$ iteration is $\alpha_t = \alpha/\sqrt{t}$, where $\alpha$ is the initial learning rate. However, in practice,

we do not use such a learning rate scheduler (*i.e.*, $\alpha/\sqrt{t}$). More advanced learning rate schedulers, *e.g.*, step-wise decay scheduler and Cosine decay scheduler, are chosen in the training process. Similar settings for learning rate are also found in the Regret-bound analysis of other Adam-type optimizers, such as RAdam Liu et al. (2019a), Adabelief Zhuang et al. (2020). People usually ignore such inconsistency between the learning rate schedulers in Regret-bound analysis and in practice for Adam-type optimizers, because of the large improvements in performance.

Nevertheless, for the AdaGrad-type optimizers, the Regret-bound analysis usually does not assume such a learning rate scheduler (*i.e.*, $\alpha/\sqrt{t}$). For example

**Theorem 1 Duchi et al. (2011); Gupta et al. (2018).** *Let $\{\boldsymbol{w}_t\}_{t=1}^T$ and $\{\boldsymbol{h}_t\}_{t=1}^T$ be the sequences obtained from AdaGrad algorithm, Which follows*

$$\boldsymbol{h}_T = \sum\nolimits_{t=1}^{T} \boldsymbol{g}_t \odot \boldsymbol{g}_t, \;\; \boldsymbol{w}_T = \boldsymbol{w}_{T-1} - \alpha \boldsymbol{h}_T^{\odot - \frac{1}{2}} \odot \boldsymbol{g}_T, \;\; T = 1, 2, 3..., \tag{6}$$

*and $\alpha$ is the learning rate, if we further assume $D = \max_{t \leq T} ||\boldsymbol{w}_t - \boldsymbol{w}^*||_2$, then we have the following bound on the regret*

$$R(T) \leq (\frac{D^2}{2\alpha} + \alpha) \sum\nolimits_{i=1}^{d} ||\boldsymbol{g}_{1:T,i}||_2. \tag{7}$$

However, although there is no assumption on the learning rate schedule of the Regret-bound Theorem for AdaGrad, the effective learning rate also reduces monotonically. That is because there is an implicit learning rate schedule in AdaGrad, which is also the same as Adam, *i.e.*, $\alpha_t = \alpha/\sqrt{t}$. Since we can rewrite Theorem 1 as follows:

**Theorem 2** *Let $\{\boldsymbol{w}_t\}_{t=1}^T$ and $\{\boldsymbol{h}_t'\}_{t=1}^T$ be the sequences obtained from*

$$\boldsymbol{h}_T' = \frac{1}{T} \sum\nolimits_{t=1}^{T} \boldsymbol{g}_t \odot \boldsymbol{g}_t, \;\; \boldsymbol{w}_T = \boldsymbol{w}_{T-1} - \alpha_T \boldsymbol{h}_T'^{\odot - \frac{1}{2}} \odot \boldsymbol{g}_T, \;\; T = 1, 2, 3..., \tag{8}$$

*where $\alpha_T = \alpha/\sqrt{T}$ is the learning rate and $\alpha$ is the initial learning rate, if we further assume $D = \max_{t \leq T} ||\boldsymbol{w}_t - \boldsymbol{w}^*||_2$, then we have the following bound on the regret*

$$R(T) \leq (\frac{D^2}{2\alpha} + \alpha) \sum\nolimits_{i=1}^{d} ||\boldsymbol{g}_{1:T,i}||_2. \tag{9}$$

We can see that the updating formula in **Theorem 1** and **Theorem 2** is equivalent, so they have the same regret bound. In **Theorem 2**, we can find that $\boldsymbol{h}_T'$ is the average of the second-order statistic of gradients, therefore, its amplitude is stable in training, unlike $\boldsymbol{h}_T$ in **Theorem 1** which increases monotonically. Because if assume $||\boldsymbol{g}_t^{\odot 2}||_\infty \leq D_\infty$ for any $t = 1, 2, ..., T$, where $||\cdot||_\infty$ is the infinite norm, we have $||\boldsymbol{h}_T'||_\infty \leq D_\infty$, then $\boldsymbol{h}_{i,T}'^{-\frac{1}{2}} \geq \frac{1}{\sqrt{D_\infty}}$ for any $i = 1, 2, ..., d$. The effective learning rate $\boldsymbol{h}_T'^{\odot - \frac{1}{2}}$ has a low-bound and will not decrease to zero. Hence, with the rewriting formulation in Eq. (8), we can separate out the stable updates which are $\boldsymbol{h}_T'^{\odot - \frac{1}{2}} \odot \boldsymbol{g}_T$ and an explicit learning rate schedule $\alpha_T = \alpha/\sqrt{T}$.

From the above observation, we investigate that AdaGrad implicitly adopts a learning rate schedule $\alpha_T = \alpha/\sqrt{T}$, which is the same as the assumption on the learning rate of Adam in its Regret-bound analysis. However, in practice, when we utilize AdaGrad optimizer, we usually directly introduce an additional learning rate scheduler (*e.g.*, step-wise decay scheduler and Cosine decay scheduler). This means there are usually two learning rate schedulers with AdaGrad optimizers, Such two learning rate schedulers will make the learning rate decrease so fast that the effective learning rate will be too small. That is the reason why AdaGrad usually performs very badly in many tasks. In contrast, Adam usually removes the learning rate schedule $\alpha_T = \alpha/\sqrt{T}$ in theoretical analysis and only applies one advanced learning rate scheduler for training in real applications. Therefore, compared with Adam, it is not fair for AdaGrad with two learning rate schedulers in practice. We think the most common utilization of AdaGrad in training DNNs is improper and it largely limits its potential performance.

### 3.3 LEARNING RATE RE-SCHEDULING FOR THE ADAGRAD

Because there are two learning rate schedulers when using AdaGrad for training DNNs, we need to remove one of them to avoid the effective learning decreasing so fast. Similar to Adam, we also attempt to eliminate the learning rate schedule $\alpha_T = \alpha/\sqrt{T}$ and only adopt one advanced learning rate scheduler (*e.g.*, step-wise decay scheduler or Cosine decay scheduler). We name the proposed

**Algorithm 1:** AdaGrad

**Input:** $\boldsymbol{w}_0, \epsilon, \boldsymbol{h}_0 = \epsilon\mathbf{1}, \alpha, \lambda, f(\alpha, \cdot)$
**Output:** $\boldsymbol{w}_T$
1 **for** *t=1:T* **do**
2     Receive $\boldsymbol{g}_t$ by backward propagation;
3     Add weight decay: $\hat{\boldsymbol{g}}_t = \boldsymbol{g}_t + \lambda\boldsymbol{w}_{t-1}$;
4     Update statistic:
5        $\boldsymbol{h}_t = \boldsymbol{h}_{t-1} + \hat{\boldsymbol{g}}_t \odot \hat{\boldsymbol{g}}_t$;
6     Compute learning rate:
7        $\alpha_t = f(\alpha, t)$;
8     Update weight:
9        $\boldsymbol{w}_t = \boldsymbol{w}_{t-1} - \alpha_t \boldsymbol{h}_t^{\odot - \frac{1}{2}} \odot \boldsymbol{g}_t$ ;
10 **end**

**Algorithm 2:** AdaGradW (Ours)

**Input:** $\boldsymbol{w}_0, \epsilon, \boldsymbol{h}_0 = \epsilon\mathbf{1}, \alpha, \lambda, f(\alpha, \cdot)$
**Output:** $\boldsymbol{w}_T$
1 **for** *t=1:T* **do**
2     Receive $\boldsymbol{g}_t$ by backward propagation;
3     Update statistic:
4        $\boldsymbol{h}_t = \boldsymbol{h}_{t-1} + \boldsymbol{g}_t \odot \boldsymbol{g}_t$;
5     Compute learning rate:
6        $\alpha_t = f(\alpha, t)$;
7     Update weight and add weight decay:
       $\boldsymbol{w}_t = \boldsymbol{w}_{t-1} - \alpha_t(\sqrt{t}\boldsymbol{h}_t^{\odot - \frac{1}{2}} \odot \boldsymbol{g}_t + \lambda\boldsymbol{w}_{t-1})$;
8 **end**

optimizer as Learning Rate Re-scheduling (LRR), whose formulation is

$$\boldsymbol{w}_T = \boldsymbol{w}_{T-1} - \sqrt{T}\alpha_T \boldsymbol{h}_T^{\odot - \frac{1}{2}} \odot \boldsymbol{g}_T, \tag{10}$$

Where $\alpha_T$ is the learning rate of the $T$-th iteration with a learning rate scheduler, $\boldsymbol{h}_T$ is defined in Eq. (3). We only change is the learning rate in the formulation of AdaGrad from $\alpha_T$ to $\sqrt{T}\alpha_T$, so the proposed method is very simple and very easy to implement in the algorithm. The additional learning rate multiplier $\sqrt{T}$ is used for eliminating the implicit learning rate schedule in AdaGrad.

Meanwhile, as mentioned in Section 3.1 it is difficult for the original AdaGrad to introduce a proper weight decay method. Therefore, the generalization performance of AdaGrad is usually unfavorable, when compared with Adam-type optimizers. Fortunately, with the proposed LRR for AdaGrad, the decoupled weight decay can be easily introduced into the optimizers, and the formulation is

$$\boldsymbol{w}_T = \boldsymbol{w}_{T-1} - \alpha_T(\sqrt{T}\boldsymbol{h}_T^{\odot - \frac{1}{2}} \odot \boldsymbol{g}_T + \lambda\boldsymbol{w}_{T-1}). \tag{11}$$

Importantly, the efficient updating term $\sqrt{T}\boldsymbol{h}_T^{\odot - \frac{1}{2}} \odot \boldsymbol{g}_T$ is usually more stable. Unlike the primitive AdaGrad algorithm, its amplitude is relatively fixed, compared with the second term $\lambda\boldsymbol{w}_T$. As a result, the decoupled weight decay can perform well with the proposed LRR operation on AdaGrad. The final generalization performance can be largely improved by this decoupled weight decay regularization.

The proposed optimizer is termed AdaGradW. **Algorithm 1** and **Algorithm 2** show the implementation of the original AdaGrad algorithm and the proposed AdaGradW algorithm. We use $f(\alpha, \cdot)$ to denote a learning rate scheduler with initial learning rate $\alpha$, and $f(\alpha, t)$ is the learning rate obtained by the learning rate scheduler in $t$-th iteration. The AdaGradW algorithm adopts the proposed learning rate re-scheduling operation and the decoupled weight decay regularization.

### 3.4 EXTENSION ON MORE ADAGRAD-TYPE OPTIMIZERS

In order to achieve a lower regret-bound, Duchi et al. (2011) also proposed a full-matrix preconditioned gradient descent (PGD) method that uses the matrix $\boldsymbol{H}_T = (\sum_{t=1}^T \boldsymbol{g}_t\boldsymbol{g}_t^\top)^{\frac{1}{2}}$ to modify the gradient. However, such formulation of the full matrix requires very high computation and memory costs, which makes it very difficult to use for training DNNs due to high dimension parameter space. Shampoo Gupta et al. (2018) and AdaBK Yong et al. (2023) were proposed to apply the Kronecker factorization on full matrix $\boldsymbol{H}_T$ to reduce its dimension. However, directly adopting such algorithms usually does not perform well. Because they are also AdaGrad-type optimizers, they also suffer from the two learning rate schedulers problem like AdaGrad. Yong et al. (2023) proposed a series of tricks to make AdaBK perform favorably, including gradient norm recovery, adaptive dampening, momentum for statistics, and embedding it into SGDM and Adam. such tricks make the proposed optimizers far from the theoretically derived optimization algorithms. Meanwhile, for Shampoo optimizer, as far as we know, there is no previous research that makes it perform comparable with Adam-type optimizers. Most of them can only outperform the original AdaGrad optimizers. Anil et al. (2020) proposed to use a norm recovery operation to change the update stepsize of shampoo to the AdaGrad. But its performance is also unsatisfactory. Here we also introduce the proposed learning rate re-scheduling operation into the Shampoo optimization algorithm. And we find that it can also boost the final performance with a large improvement. **Algorithm 3** and **Algorithm 4** summary the original Shampoo optimizer and the proposed ShampooW optimizer.

---

**Algorithm 3:** Shampoo

**Input:** $\boldsymbol{W}_0 \in \mathbb{R}^{C_{out} \times C_{in}}$, $\epsilon$, $\boldsymbol{L}_0 = \epsilon \boldsymbol{I}_{C_{out}}$,
$\boldsymbol{R}_0 = \epsilon \boldsymbol{I}_{C_{in}}$, $\alpha$, $\lambda$, $f(\alpha, \cdot)$, $T_s$, $T_{ir}$
**Output:** $\boldsymbol{W}_T$

1 **for** *t=1:T* **do**
2     Receive $\boldsymbol{G}_t$ by backward propagation;
3     Add weight decay: $\hat{\boldsymbol{G}}_t = \boldsymbol{G}_t + \lambda \boldsymbol{W}_{t-1}$;
4     **if** $t\%T_s = 0$ **then**
5       $\boldsymbol{L}_t = \boldsymbol{L}_{t-1} + \hat{\boldsymbol{G}}_t \hat{\boldsymbol{G}}_t^\top$ ;
6       $\boldsymbol{R}_t = \boldsymbol{R}_{t-1} + \hat{\boldsymbol{G}}_t^\top \hat{\boldsymbol{G}}_t$;
7     **else**
8       $\boldsymbol{L}_t = \boldsymbol{L}_{t-1}, \boldsymbol{R}_t = \boldsymbol{R}_{t-1}$;
9     **end**
10     **if** $t\%T_{ir} = 0$ **then**
11       $\boldsymbol{U}_1 \boldsymbol{\Sigma}_1 \boldsymbol{U}_1 = \boldsymbol{L}_t, \widehat{\boldsymbol{L}}_t = \boldsymbol{U}_1 \boldsymbol{\Sigma}_1^{-\frac{1}{4}} \boldsymbol{U}_1$
12       $\boldsymbol{U}_2 \boldsymbol{\Sigma}_2 \boldsymbol{U}_2 = \boldsymbol{R}_t, \widehat{\boldsymbol{R}}_t = \boldsymbol{U}_2 \boldsymbol{\Sigma}_2^{-\frac{1}{4}} \boldsymbol{U}_2$
13     **else**
14       $\widehat{\boldsymbol{L}}_t = \widehat{\boldsymbol{L}}_{t-1}$ and $\widehat{\boldsymbol{R}}_t = \widehat{\boldsymbol{L}}_{t-1}$;
15     **end**
16     Compute learning rate:
17       $\alpha_t = f(\alpha, t)$;
18     Update weight: $\boldsymbol{W}_t = \boldsymbol{W}_{t-1} - \alpha_t \widehat{\boldsymbol{L}}_t \hat{\boldsymbol{G}}_t \widehat{\boldsymbol{R}}_t$;
19 **end**

---

**Algorithm 4:** ShampooW (Ours)

**Input:** $\boldsymbol{W}_0 \in \mathbb{R}^{C_{out} \times C_{in}}$, $\epsilon$, $\boldsymbol{L}_0 = \epsilon \boldsymbol{I}_{C_{out}}$,
$\boldsymbol{R}_0 = \epsilon \boldsymbol{I}_{C_{in}}$, $\alpha$, $\lambda$, $f(\alpha, \cdot)$, $T_s$, $T_{ir}$
**Output:** $\boldsymbol{W}_T$

1 **for** *t=1:T* **do**
2     Receive $\boldsymbol{G}_t$ by backward propagation;
3     **if** $t\%T_s = 0$ **then**
4       $\boldsymbol{L}_t = \boldsymbol{L}_{t-1} + \boldsymbol{G}_t \boldsymbol{G}_t^\top$ ;
5       $\boldsymbol{R}_t = \boldsymbol{R}_{t-1} + \boldsymbol{G}_t^\top \boldsymbol{G}_t$;
6     **else**
7       $\boldsymbol{L}_t = \boldsymbol{L}_{t-1}, \boldsymbol{R}_t = \boldsymbol{R}_{t-1}$;
8     **end**
9     **if** $t\%T_{ir} = 0$ **then**
10       $\boldsymbol{U}_1 \boldsymbol{\Sigma}_1 \boldsymbol{U}_1 = \boldsymbol{L}_t, \widehat{\boldsymbol{L}}_t = \boldsymbol{U}_1 \boldsymbol{\Sigma}_1^{-\frac{1}{4}} \boldsymbol{U}_1$
11       $\boldsymbol{U}_2 \boldsymbol{\Sigma}_2 \boldsymbol{U}_2 = \boldsymbol{R}_t, \widehat{\boldsymbol{R}}_t = \boldsymbol{U}_2 \boldsymbol{\Sigma}_2^{-\frac{1}{4}} \boldsymbol{U}_2$
12     **else**
13       $\widehat{\boldsymbol{L}}_t = \widehat{\boldsymbol{L}}_{t-1}$ and $\widehat{\boldsymbol{R}}_t = \widehat{\boldsymbol{L}}_{t-1}$;
14     **end**
15     Compute learning rate:
16       $\alpha_t = f(\alpha, t)$;
17     Update weight and add weight decay:
      $\boldsymbol{W}_t = \boldsymbol{W}_{t-1} - \alpha_t (\sqrt{t} \widehat{\boldsymbol{L}}_t \boldsymbol{G}_t \widehat{\boldsymbol{R}}_t + \lambda \boldsymbol{W}_{t-1})$;
18 **end**

---

## 4 EXPERIMENTAL RESULTS

We evaluate the proposed AdaGradW and ShampooW optimizers on classical computer vision tasks, including image classification (on CIFAR100/CIFAR10 Krizhevsky et al. (2009) and ImageNet Russakovsky et al. (2015)). All experiments are conducted under the Pytorch 1.18 framework with NVIDIA GeForce RTX 2080Ti and 3090 Ti GPUs. For the hyper-parameters of AdaGrad and AdaGradW, $\epsilon$ is set to be $1e^{-10}$, and for the Shampoo and ShampooW, we set $\epsilon = 1e^{-4}$, $T_s = 10$, $T_{ir} = 500$, throughout the experiments if not specified. Other hyper-parameters (*i.e.*, learning rate and weight decay) are reported in the **supplementary materials**. For all tables, the best and second-best results are highlighted in bold and italic fonts, respectively.

### 4.1 IMAGE CLASSIFICATION ON CIFAR100/CIFAR10

#### 4.1.1 EFFECTIVENESS OF LEARNING RATE RE-SCHEDULING

We first testify to the effectiveness of learning rate re-scheduling for AdaGrad. In this experiment, we adopt AdaGrad to train ResNet18, ResNet50 He et al. (2016) on CIFAR100. The four methods are varied: the original AdaGrad with L2 regularization weight decay (L2WD), AdaGrad with decoupled weight decay (DWD), AdaGrad with L2WD and the proposed learning rate rescheduling (LRR), AdaGrad with DWD and LRR. All the DNN models are trained for 200 epochs with batch size 128 on one GPU. The learning rate schedule is step-wise decay, which multiplies 0.1 on the learning rate for every 60 epochs. The experiments are repeated 4 times and the results are reported in a "mean±std" format in Table 1. Meanwhile, Figure 1 shows the training loss curves and test accuracy curves of AdaGrad on CIFAR100 with the ResNet18 model. From the table, we know that decoupled weight decay can improve the generalization performance of AdaGrad largely, with about 3.68% and 2.78% improvements on ResNet18 and ResNet50, respectively. However, from Figure 1, we know that the model has not been fully trained with decoupled weight decay in AdaGrad. The reason is that because of the implicit learning rate schedule in AdaGrad (*i.e.*, $\alpha_t = \alpha/\sqrt{t}$), the decoupled weight decay term will dominate the stepsize in certain iterations, which limits the optimization of the loss function. It can be seen from Figure 1 that the loss even increases after 150 epochs for DWD. Moreover, with the proposed learning rate rescheduling, the performance can further gain 2.59% and 4.47% over AdaGrad with DWD on ResNet18 and ResNet50, respectively. Meanwhile, the training loss can be

Table 1: Test accuracy (%) on CIFAR100 with ResNet18 and ResNet50. All models are trained with AdaGrad optimizer. L2WD: L2 regularization weight decay; DWD: decoupled weight decay; LRR: learning rate rescheduling.

| Methods | ResNet18 | ResNet50 |
|---------|----------|----------|
| L2WD | $71.55 \pm .25$ | $72.20 \pm .15$ |
| DWD | $75.23 \pm .39$ | $74.98 \pm .28$ |
| LRR+L2WD | $72.85 \pm .17$ | $73.28 \pm .23$ |
| LRR+DWD | $\mathbf{77.82 \pm .10}$ | $\mathbf{79.45 \pm .32}$ |

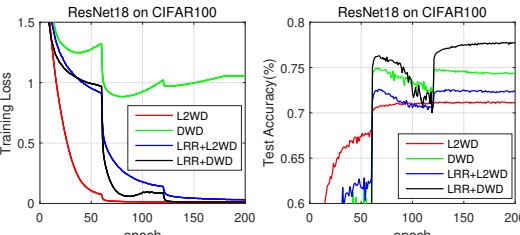

Figure 1: Training loss curves (left) and Test accuracy curves (right) of AdaGrad on CIFAR100 with ResNet18 model.

Table 2: Test accuracy (%) on CIFAR100 with different learning rate schedulers, such as Step-wise LR decay and Cosine LR decay.

| Methods | Step-wise LR decay | | | | | Cosine LR decay | | | | |
|---------|-----|-----|-----|-----|------|-----|-----|-----|-----|------|
| | R18 | R50 | V11 | V19 | D121 | R18 | R50 | V11 | V19 | D121 |
| AdaGrad | $71.55 \pm .25$ | $72.20 \pm .15$ | $67.70 \pm .18$ | $63.30 \pm .58$ | $71.27 \pm .79$ | $71.57 \pm .53$ | $72.20 \pm .63$ | $67.90 \pm .30$ | $64.32 \pm .54$ | $71.15 \pm .93$ |
| AdaGradW | $77.82 \pm .10$ | $79.45 \pm .32$ | $71.45 \pm .16$ | $71.40 \pm .36$ | $78.95 \pm .21$ | $78.67 \pm .22$ | $80.47 \pm .19$ | $72.45 \pm .32$ | $72.90 \pm .45$ | $79.47 \pm .12$ |
| Shampoo | $71.81 \pm .40$ | $71.31 \pm .53$ | $63.56 \pm .44$ | $65.62 \pm .56$ | $74.95 \pm .42$ | $72.87 \pm .75$ | $72.87 \pm .70$ | $68.22 \pm .34$ | $65.22 \pm .42$ | $71.15 \pm .77$ |
| ShampooW | $\mathbf{79.30 \pm .27}$ | $\mathbf{81.25 \pm .08}$ | $\mathbf{73.02 \pm .24}$ | $\mathbf{74.80 \pm .21}$ | $\mathbf{80.72 \pm .13}$ | $\mathbf{79.95 \pm .15}$ | $\mathbf{81.85 \pm .05}$ | $\mathbf{73.72 \pm .35}$ | $\mathbf{75.75 \pm .07}$ | $\mathbf{81.07 \pm .23}$ |

fully optimized with LLR as shown in Figure 1. It can significantly illustrate the effectiveness of the proposed learning rate re-scheduling method.

### 4.1.2 RESULTS ON DIFFERENT LEARNING RATE SCHEDULERS

To testify that the proposed AdaGradW and ShampooW can perform well on different learning rate schedule methods. In this experiment, we adopt AdaGrad, AdaGradW, Shampoo, and ShampooW to train various DNN models on CIFAR100. The DNN models include ResNet18 (R18), ResNet50 (R50) He et al. (2016), VGG11 (V11) VGG19 (V19) Simonyan & Zisserman (2014) and DenseNet-121 (D121) Huang et al. (2017) [1]. Two common learning rate schedule methods are used in this experiment, which are step-wise LR decay scheduler and Cosine LR decay scheduler. For the step-wise LR decay scheduler, we multiply 0.1 on the learning rate for every 60 epochs; while for the Cosine LR decay scheduler, the hyperparameters $T_{max} = 200$ and $eta_{min} = 0.001 * \alpha$, where $\alpha$ is the initial learning rate. The experiments are repeated 4 times and the results are reported in a "mean±std" format in Table 2. It can be seen from the figure that the Cosine LR decay scheduler usually outperforms the Step-wise LR decay scheduler, and the proposed AdaGradW and ShampooW significantly outperform AdaGrad and Shampoo on these two learning rate schedulers, respectively. It demonstrates that the proposed AdaGradW and ShampooW can work consistently well on different learning rate schedule methods.

### 4.1.3 COMPARISON WITH DIFFERENT OPTIMIZERS

we compare the proposed AdaGradW and ShampooW with some representative DNN optimizers, including including SGDM, AdamW Loshchilov & Hutter (2017), Adagrad Duchi et al. (2011), RAdam Liu et al. (2019b)[2], and Adabelief Zhuang et al. (2020)[3], Shampoo Gupta et al. (2018)[4]. The DNN models also include ResNet18 (R18), ResNet50 (R50) VGG11 (V11) VGG19 (V19) and DenseNet-121 (D121). All the DNN models are trained for 200 epochs with batch size 128 on one GPU. The learning rate schedule is step-wise decay, which multiplies 0.1 on the learning rate for every 60 epochs. The experiments are repeated 4 times and the results are reported in a "mean±std" format in Table 3. We can see that AdaGradW and ShampooW achieve significant improvements over AdaGrad and Shampoo, which are $3.75\% \sim 8.1\%$ and $7.49\% \sim 9.94\%$ on CIFAR100, and $2.2\% \sim 2.85\%$ and $1.48\% \sim 3.89\%$ on CIFAR10, respectively. They also outperform other compared optimizers for most of the used DNN models.

---

[1] The model can be downloaded at `https://github.com/weiaicunzai/pytorch-cifar100`.

[2] `https://github.com/LiyuanLucasLiu/RAdam`

[3] `https://github.com/juntang-zhuang/Adabelief-Optimizer`

[4] `https://github.com/moskomule/shampoo.pytorch`

Table 3: Testing accuracies (%) on CIFAR100/CIFAR10 with various optimizers.

| Optimizer | SGDM | AdamW | RAdam | Adabelief | AdaGrad | Shampoo | AdaGradW | ShampooW |
|---|---|---|---|---|---|---|---|---|
| | | | | CIFAR100 | | | | |
| ResNet18 | $77.20 \pm .30$ | $77.23 \pm .10$ | $77.05 \pm .15$ | $77.43 \pm .36$ | $71.55 \pm .25$ | $71.81 \pm .40$ | $77.82 \pm .10$ | $\mathbf{79.30} \pm .27$ |
| ResNet50 | $77.78 \pm .43$ | $78.10 \pm .17$ | $78.20 \pm .15$ | $79.08 \pm .23$ | $72.20 \pm .15$ | $71.31 \pm .53$ | $79.45 \pm .32$ | $\mathbf{81.25} \pm .08$ |
| VGG11 | $70.80 \pm .29$ | $71.20 \pm .29$ | $71.08 \pm .24$ | $72.45 \pm .16$ | $67.70 \pm .18$ | $63.56 \pm .44$ | $71.45 \pm .16$ | $\mathbf{73.02} \pm .24$ |
| VGG19 | $70.94 \pm .32$ | $70.26 \pm .23$ | $73.01 \pm .20$ | $72.39 \pm .27$ | $63.30 \pm .58$ | $65.62 \pm .56$ | $71.40 \pm .36$ | $\mathbf{74.80} \pm .21$ |
| DenseNet121 | $79.53 \pm .19$ | $78.05 \pm .26$ | $78.65 \pm .05$ | $79.88 \pm .08$ | $71.27 \pm .79$ | $74.95 \pm .42$ | $78.95 \pm .21$ | $\mathbf{80.72} \pm .13$ |
| | | | | CIFAR10 | | | | |
| ResNet18 | $95.10 \pm .07$ | $94.80 \pm .10$ | $94.70 \pm .18$ | $95.12 \pm .14$ | $92.83 \pm .12$ | $92.94 \pm .27$ | $95.17 \pm .12$ | $\mathbf{95.50} \pm .13$ |
| ResNet50 | $94.75 \pm .30$ | $94.72 \pm .10$ | $94.72 \pm .10$ | $95.35 \pm .05$ | $92.55 \pm .39$ | $92.61 \pm .27$ | $95.40 \pm .07$ | $\mathbf{95.82} \pm .09$ |
| VGG11 | $92.17 \pm .19$ | $92.02 \pm .08$ | $92.00 \pm .18$ | $92.45 \pm .18$ | $90.25 \pm .25$ | $89.01 \pm .29$ | $92.45 \pm .11$ | $\mathbf{92.90} \pm .17$ |
| VGG19 | $93.61 \pm .06$ | $93.40 \pm .04$ | $93.57 \pm .11$ | $93.58 \pm .12$ | $91.28 \pm .14$ | $90.62 \pm .32$ | $93.82 \pm .10$ | $\mathbf{94.12} \pm .14$ |
| DenseNet121 | $95.37 \pm .17$ | $94.80 \pm .07$ | $95.02 \pm .08$ | $95.37 \pm .04$ | $92.95 \pm .23$ | $94.37 \pm .36$ | $95.17 \pm .10$ | $\mathbf{95.85} \pm .11$ |

Table 4: Top 1 accuracy (%) of various optimizers on the validation set of ImageNet-1k.

| Optimizer | SGDM | AdamW | RAdam | Adabelief | AdaGrad | Shampoo | AdaGradW | ShampooW |
|---|---|---|---|---|---|---|---|---|
| ResNet18 | $70.49$ | $70.01$ | $69.92$ | $70.08$ | $62.22$ | $64.45$ | $70.26$ | $\mathbf{71.32}$ |
| ResNet50 | $76.31$ | $76.02$ | $76.12$ | $76.22$ | $69.38$ | $70.11$ | $76.52$ | $\mathbf{77.02}$ |

## 4.2 IMAGE CLASSIFICATION ON IMAGENET1K

We also evaluate AdaGradW and ShampooW on ImageNet-1kRussakovsky et al. (2015) to testify that the proposed LRR can also perform well on large-scale datasets. It contains 1000 categories with 1.28 million images for training and 50K images for validation. ResNet18 and ResNet50 are selected as the DNN models with training batch size 256 on 4 GPUs, and the training settings follow the work in Chen et al. (2018); Zhuang et al. (2020); Yong et al. (2023). Step-wise LR decay scheduler is used by multiplying $0.1$ on the learning rate for every 30 epochs. The top 1 accuracies on the validation set are reported in Table 4. One can see that AdaGradW and ShampooW achieve favorable performance and considerable improvements over AdaGrad and Shampoo, about $7.14\% \sim 8.04\%$ and $6.87\% \sim 6.91\%$. respectively.

Table 5: Top 1 accuracy (%) on the validation set of ImageNet-1k with Swin-transformer backbone.

| Optimizer | AdamW | AdaGradW |
|---|---|---|
| Swin-T | 81.18 | 81.19 |
| Swin-B | 83.02 | 83.00 |

Finally, we compare AdaGradW with the default optimizer AdamW on Swin-transformer Liu et al. (2021) backbone. The configurations follow the settings of the official mm-pretrain toolbox[5]. The results are shown in Table 5 give the result. AdaGradW can also achieve comparable performance over AdamW.

## 5 CONCLUSION

This work points out the problems of the well-known optimization algorithm AdaGrad in training DNNs. The first problem is the effective learning rate decreases during all the training, which makes the weights change very slowly after certain iterations. Another problem is the decoupled weight decay regularization will have a negative impact on the optimization of the loss function. The reason for such problems of AdaGrad in training DNNs is the incorrect usage in practice. People usually ignore the implicit learning rate schedule in AdaGrad. However, an explicit learning rate scheduler and such implicit learning rate scheduler at the same time for AdaGrad will lead to an unfavorable performance in training DNNs. We propose a learning rate re-scheduling (LRR) method to address this problem by eliminating the implicit learning rate scheduler in AdaGrad. Meanwhile, the decoupled weight decay can also be perfectly added to the AdaGrad optimizer with LRR. Moreover, the proposed LRR can also be introduced into other AdaGrad-type optimizers, such as Shampoo. The proposed optimizers are named AdaGradW and ShampooW, and their effectiveness is illustrated by sufficient experimental results on image classification tasks.

---

[5] https://github.com/open-mmlab/mmpretrain/tree/master/configs/swin_transformer

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
