# Supplementary Materials to: "Learning Rate Re-scheduling for AdaGrad in training Deep Neural Networks"

## Abstract

The hyper-parameter settings of different optimizers, including learning rate and weight decay, are provided in this supplementary file.

## A. The Hyper-parameter Settings on CIFAR100/CIFAR10

We First introduce the hyper-parameters of different optimizers on CIFAR100/CIFAR10. We tune the LR and WD of all optimizers by grid search. The learning rate is tunned from $\{1e^{-4}, 5e^{-4}, 1e^{-3}, 5e^{-3}, 1e^{-2}, 5e^{-2}, 0.1\}$ and weight decay is tuned from $\{1e^{-4}, 5e^{-4}, 1e^{-3}, 5e^{-3}, 1e^{-2}, 5e^{-2}, 0.1, 0.5\}$. Then we choose the best combination of learning rate and weight decay for all optimizers. Table 1 shows the final hyper-parameter settings for different AdaGrad methods *cf.* of Section 4.1.1 in the main paper. Meanwhile, Table 2 shows the final hyper-parameter settings for different optimizers *cf.* of Section 4.1.2 and 4.1.3 in the main paper. Other hyper-parameters are set to be the default settings.

Table 1: Settings of learning rate (LR), weight decay (WD) for different AdaGrad Methods on CIFAR10/100.

| Methods | LR | WD |
|---------|------|--------|
| L2WD | 0.01 | 0.0005 |
| DWD | 0.01 | 0.05 |
| LRR+L2WD | 0.001 | 0.0005 |
| LRR+DWD | 0.001 | 0.5 |

Table 2: Settings of learning rate (LR), weight decay (WD) and WD methods for different optimizers on CIFAR10/100. Here, the WD methods include $L_2$ regularization weight decay ($L_2$ in short) and weight decouple (decouple in short).

| Optimizer | SGDM | AdamW | RAdam | Adabelief | Adagrad | Shampoo | AdaGradW | ShampooW |
|-----------|------|-------|-------|-----------|---------|---------|----------|----------|
| LR | 0.1 | 0.001 | 0.001 | 0.001 | 0.01 | 0.001 | 0.001 | 0.001 |
| WD | 0.0005 | 0.5 | 0.5 | 0.5 | 0.0005 | 0.0005 | 0.5 | 0.5 |
| WD method | $L_2$ | decouple | decouple | decouple | $L_2$ | $L_2$ | decouple | decouple |

## B. The Hyper-parameter Settings on ImageNet

We then introduce the hyper-parameters of different optimizers on ImageNet. We refer to the strategies in Zhuang et al. (2020) to tune the LR and WD on ResNet18 and ResNet50, respectively. The final settings are described in Table 3 *cf.* of Section 4.2 in the main paper. Other hyper-parameters are set to be the default settings. Meanwhile, we plot the training and validation accuracy curves in Figure 1, from which we see that the proposed LRR technique can largely improve the final performance.

Table 3: Settings of learning rate (LR), weight decay (WD) and WD methods ($L_2$ and decouple) for different optimizers on ImageNet.

| Optimizer | | SGDM | AdamW | RAdam | Adabelief | Adagrad | Shampoo | AdagradW | ShampooW |
|---|---|---|---|---|---|---|---|---|---|
| ResNet18 | LR | 0.1 | 0.001 | 0.001 | 0.001 | 0.01 | 0.001 | 0.001 | 0.001 |
| | WD | 0.0001 | 0.1 | 0.1 | 0.05 | 0.0001 | 0.0001 | 0.1 | 0.1 |
| ResNet50 | LR | 0.1 | 0.001 | 0.001 | 0.001 | 0.01 | 0.001 | 0.001 | 0.001 |
| | WD | 0.0001 | 0.1 | 0.05 | 0.1 | 0.0001 | 0.0001 | 0.1 | 0.1 |
| WD method | | $L_2$ | decouple | decouple | decouple | $L_2$ | $L_2$ | decouple | decouple |

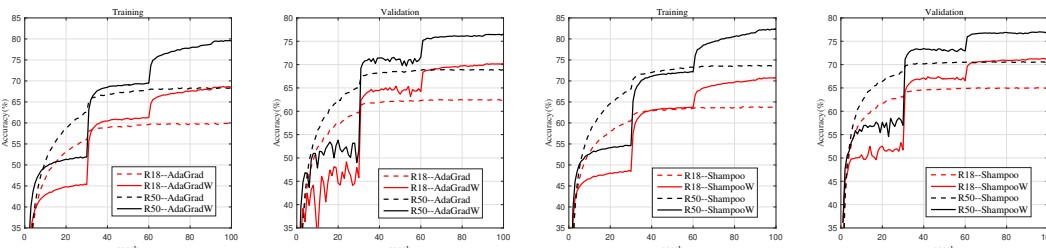

Figure 1: Training and validation accuracy curves of AdaGrad, AdaGradW, Shampoo and ShampooW on ImageNet-1k with ResNet18 and ResNet50 backbones.

## REFERENCES

Juntang Zhuang, Tommy Tang, Yifan Ding, Sekhar Tatikonda, Nicha Dvornek, Xenophon Papademetris, and James S Duncan. Adabelief optimizer: Adapting stepsizes by the belief in observed gradients. *arXiv preprint arXiv:2010.07468*, 2020.