# OpenReview forum: "Learning Rate Re-scheduling for AdaGrad in training Deep Neural Networks"
_ICLR.cc/2024/Conference — ICLR 2024 Conference Withdrawn Submission_

### Official Review · Reviewer_mVqq · 2023-10-30

**Soundness:** 2 fair
**Presentation:** 2 fair
**Contribution:** 3 good
**Rating:** 3
**Confidence:** 4

**Summary:**

The paper addresses the challenges faced when using the AdaGrad optimizer in training Deep Neural Networks (DNNs). While adaptive learning rate optimization algorithms have significantly improved DNN training, AdaGrad often underperforms compared to newer optimizers like Adam, RAdam, and Adabelief. The paper identifies two primary reasons for AdaGrad's limitations: the decreasing updating delta and the unfavorable performance of decoupled weight decay regularization. The authors highlight a common mistake in using AdaGrad with an additional learning rate schedule, which negatively impacts its performance. To address this, the paper introduces a Learning Rate Re-scheduling (LRR) method for AdaGrad, aiming to improve its performance and make decoupled weight decay regularization more effective. The LRR method can also be applied to other AdaGrad-type algorithms, and experimental results support its effectiveness.

**Strengths:**

1. The paper clearly outlines the challenges of using AdaGrad in DNN training, providing a foundation for their proposed solution. The introduction of the Learning Rate Re-scheduling method offers a fresh perspective on improving AdaGrad's performance.
2. The LRR method's compatibility with other AdaGrad-type algorithms increases its potential impact in the deep learning community. The performance improvement over the baseline adagrad and shampoo looks impressive.

**Weaknesses:**

My major concerns are around the motivation and reproducibility of this work. I would raise the score if all problems are addressed.

1. The paper assumes that most practitioners add an additional learning rate schedule to AdaGrad, which might not be universally true.
2. The performance over other more popular optimizers (e.g., SGD and ADAM), are not significant. Thus motivation should be more solid, why we need a new adagrad over the other choices.
3. Reproducibility: There's no code provided in the supplemental materials, which is crucial for evaluating this paper that emphasize "practice" for multiple times and targets for higher performance.

**Questions:**

1. Could the authors provide the original code in an anonymous link for reproducibility? It's better to have an easy-to-run toy example for showing the effectiveness of proposed method.

---

### Official Review · Reviewer_fweP · 2023-11-01

**Soundness:** 3 good
**Presentation:** 3 good
**Contribution:** 1 poor
**Rating:** 3
**Confidence:** 4

**Summary:**

This paper proposes a learning rate schedule for AdaGrad, which allows it to match the performance of AdamW when the weight decay is decoupled. The same method is also applied to Shampoo, which allows it to surpass the performance of the original Shampoo and also AdamW for ResNet, VGG, and DenseNet on Cifar100.

**Strengths:**

This work addresses a major issue with the original AdaGrad method and provides a fix that makes sense. Experiments on five different CNNs show a consistent improvement of 5-10% in test accuracy for both AdaGrad and Shampoo, which is significant. The description of their methods and motivation are clear.

**Weaknesses:**

The work by Anil et al. [https://arxiv.org/abs/2002.09018] which the authors cite, actually uses a moving average for  L and R, though this is not clear from the equations in the paper. In the appendix, Algorithm II shows that they use a moving average. This fixes the sqrt(t) issue addressed in the current work. They also use the decoupled weight decay, and should essentially achieve the same performance as the ShampooW proposed in the current work. This paper in 2020 is referenced by some more recent work on Shampoo, which follow the same practice. In light of this previous work, it doesn't seem like the proposed method adds any practical benefit to the existing versions of Shampoo. Also, there is no strong reason to believe that AdaGradW adds any practical benefit over AdamW as well.

**Questions:**

What is the authors' position on these version of Shampoo that use a moving average for L and R?
Why are there no experiments for language tasks, which are known to benefit more from AdamW?

---

### Official Review · Reviewer_49a3 · 2023-11-04

**Soundness:** 2 fair
**Presentation:** 2 fair
**Contribution:** 2 fair
**Rating:** 3
**Confidence:** 3

**Summary:**

This paper proposes a learning rate re-scheduling method for AdaGrad. Moreover, the authors also extend the proposed method on another AdaGrad-type optimizer, Shampo. Some experimental results show the performance of the proposed method.

**Strengths:**

1. This paper proposes a learning rate re-scheduling method for AdaGrad.
2. Moreover, the authors also extend the proposed method on another AdaGrad-type optimizer, Shampo.
3. Some experimental results show the performance of the proposed method.

**Weaknesses:**

Although the paper is theoretically and experimental sound, there are still some questions need to be discussed in this paper:
1.	The main contribution of this paper is to propose a learning rate re-scheduling method. But the detail of the proposed learning rate re-scheduling method is missing.
In particular, the authors should compare the proposed learning rate re-scheduling method with existing methods.
2.	In Eq. (3), where is the element-wise matrix product used? Note that g_t should be a vector.
3.	Is \sqrt{T} a constant?
4.	The convergence analysis of the variants of AdaGrad and Shampo should be provided.
5.	The experimental results are not convincing. The authors should compare the proposed algorithm with more recently proposed algorithms.
6.	Both the English language and equations in this paper need to be improved.

**Questions:**

Although the paper is theoretically and experimental sound, there are still some questions need to be discussed in this paper:
1.	The main contribution of this paper is to propose a learning rate re-scheduling method. But the detail of the proposed learning rate re-scheduling method is missing.
In particular, the authors should compare the proposed learning rate re-scheduling method with existing methods.
2.	In Eq. (3), where is the element-wise matrix product used? Note that g_t should be a vector.
3.	Is \sqrt{T} a constant?
4.	The convergence analysis of the variants of AdaGrad and Shampo should be provided.
5.	The experimental results are not convincing. The authors should compare the proposed algorithm with more recently proposed algorithms.
6.	Both the English language and equations in this paper need to be improved.

---

### Meta-Review · Area_Chair_mLHq · 2023-12-24

**Metareview:**

The paper lacks detailed comparison of the proposed learning rate re-scheduling method with existing ones. The experiments need to include more recent algorithms, and the paper requires linguistic and mathematical refinements. There are also concerns regarding the novelty and practical contributions of the proposed methods in comparison to prior works, especially by Anil et al., which seems to cover similar ground. The lack of supplemental materials, such as code, limits the paper's reproducibility. All reviewers are not supportive, and there is no response from the authors. I recommend reject.

**Justification For Why Not Higher Score:**

All reviewers vote for reject. No response from authors.

**Justification For Why Not Lower Score:**

N/A

---

### Decision · Program_Chairs · 2024-01-16

Reject